# European Educational Programmes in Health Emergency and Disaster Management: An Integrative Review

**DOI:** 10.3390/ijerph182111455

**Published:** 2021-10-30

**Authors:** Juana Perpiñá-Galvañ, Rocío Juliá-Sanchis, Érika Olmos-Castelló, Salvador Mollá-Pérez, Ángela Sanjuan-Quiles

**Affiliations:** 1Department of Nursing, University of Alicante, 03690 Alicante, Spain; juana.perpina@ua.es (J.P.-G.); eolcas@alumni.uv.es (É.O.-C.); salva.molla@ua.es (S.M.-P.); angela.sanjuan@ua.es (Á.S.-Q.); 2Alicante Institute for Health and Biomedical Research (ISABIAL), 03690 Alicante, Spain

**Keywords:** educational programme, emergencies, European, disasters

## Abstract

There is a need for trained health professionals who can swiftly respond to disasters occurring worldwide. Little is known about whether the currently available programmes in disaster management are in line with the recommendations of expert researchers. Our objective was to qualitatively review the characteristics of European educational programmes in health emergency and disaster management and to provide guidance to help improve their curricula. We carried out an integrative review to extract the main characteristics of the 2020/21 programmes available. We identified 34 programmes, the majority located in Spain, the UK or France. The primary qualification types awarded were master’s degrees, half of them lasting one year, and the most common teaching method was in person. Almost all of the programmes used a virtual university classroom, a third offered multidisciplinary disaster management content and teachers, and half of them employed situational simulations. The quality of European educational programmes in health emergency and disaster management has improved, especially in terms of using more practical and interactive teaching methodologies and in the inclusion of relevant topics such as communication, psychological approaches and evaluation of the interventions. However, generally, the educational programmes in disaster management have not yet incorporated the skills related to the intercultural and interprofessional awareness aspects.

## 1. Introduction

In recent years, emergencies such as natural disasters (earthquakes, tsunamis, floods or fires), the outbreak of the SARS-CoV-2 pandemic, intercontinental travel of migrant and refugee populations, and other human-made disasters (armed conflicts, terrorist attacks, or nuclear or chemical technological failures) have had a worldwide effect [1,2]. Situations are considered disasters when the event exceeds the capacity of a country or community to cope with it, thus generating material, economic, environmental or human losses (between one thousand and one million) [3], and seriously disrupting the functioning of the affected society [4]. Regardless of the disaster trigger, the result is often a cascade of human suffering in the form of large-scale displacements, food shortages, disease outbreaks, violations of people’s rights and dignity, and even deaths [5]. Thus, at a global level, different disaster risk-reduction strategies have been implemented to increase nations’ and communities’ resilience to disasters and to reduce losses in human lives and in social, economic and environmental assets [3].

With the agreement of the 2015–2030 Sendai Framework for Disaster Risk Reduction, measures were established for the three dimensions of disaster risk (exposure to hazards, vulnerability and capacity, and hazard characteristics) to help prevent new risks, reduce risks and increase the resilience of communities or regions [6,7]. In addition, Marco Sendai highlighted improvement in the training of health and non-health responders as a key element in effective disaster responses [7]. Intervention teams often feel insufficiently prepared to act because they believe that their level of knowledge and technical skills are inadequate, and their psychological or organisational preparation is poor [5,8]. In addition, they believe that they have a diffuse leadership and an ambiguous distribution of roles [9] during disasters. All of this tends to lead to misunderstandings and organisational chaos, especially in terms of the command–control–communication triad [10] between the involved parties (firefighters, state security forces, architects, engineers or other healthcare professionals). This has significant negative implications for the effectiveness of interventions, responses to the complex demands of disasters and the health outcomes of the affected population [11,12]. Thus, we believe that training in disaster management should be improved [10,13,14].

A European multidisciplinary team of experts collaborated to develop a standardised curriculum for international crises management, hereby referred to as DITAC (Disaster Training Curriculum), funded by the European Union in 2014 [15]. The DITAC Project developed a holistic and standardised training plan aimed at all potential disaster responders and managers. Researchers started by reviewing 140 educational programmes in disaster management (EPDM) to search for commonalities between them. Thus, they proposed that the minimum characteristics of an EPDM were that it should be skills-based, practical and multidisciplinary, and competent to help resolve disaster scenarios. Subsequently, these researchers proposed a new programme for integrated operational, tactical and strategic training at every level of disaster coordination, communication and cooperation [16], not to re-educate professionals in their profession but, rather, to train them in disaster management with a greater intercultural and interinstitutional emphasis [10] based on the following themes: (1) the theory of intercultural cooperation, setting out the general principles of disasters, disaster cycles and their management, cultural factors, political considerations and European organisation models; (2) intercultural awareness, addressing the principles of public health, the main medical problems, search and rescue, security and protection, logistics, mental health care and psychological support, all based on an intercultural understanding; (3) interprofessional skills, especially communication skills in planning and management, leadership and decision making, information gathering and exchange, legal and ethical factors, non-governmental and voluntary organisations, risk and vulnerability analysis, recovery and reconstruction; (4) complex scenarios (holistic training) that would combine all of the above through simulations [10,15].

Without contradicting the global approach taken by the DITAC project, the 2015–30 Sendai Framework places human health at the centre of disaster management and considers the development of healthcare personnel in this area a vital issue [6]. However, since then, several publications have shown that the technical (knowledge) and methodological (know-how) skills of doctors and nurses remain insufficient in this field [9,13,16,17,18,19,20]. Despite the efforts made to standardise EPDMs over the last 7 years, we still do not know whether the European EPDMs aimed at health professionals have incorporated the evidence-based recommendations that standardised curricula should be developed to deal with the challenges posed by disasters. Based on recent literature reviews [21], we hypothesised that gaps in skills development in this area still persist in these workers. Thus, there is a need to renew European health professionals’ understanding of EPDMs and to provide directions for their further development. Therefore, the purpose of this study was to qualitatively review the characteristics of European EPDMs and to provide guidance on ways their curricula could be improved.

## 2. Materials and Methods

Three researchers conducted a qualitative synthesis of the content regarding disaster management published on European university websites, by implementing an integrative review framework [22]. This review is presented in accordance with the ENTREQ presentation criteria [23]. The ENTREQ statement helps researchers to report the stages associated with the synthesis of qualitative health research: searching and selecting qualitative research, quality appraisal and methods for synthesising qualitative findings [23]. The review protocol was not pre-registered.

### 2.1. Study Selection

The inclusion criteria for the courses considered were (1) postgraduate EPDM; (2) aimed at medical or nursing professionals; (3) available during the 2020/21 academic year; (4) information (curricula and teaching guides) that could be cross-checked on the university website or upon request by email; (5) information expressed in English, French or Spanish; (6) course delivery by a public or private university; (7) conducted in one of the following European countries (Denmark, Sweden, Norway, Finland, Netherlands, Switzerland, Luxembourg, United Kingdom, Ireland, Portugal, Spain, Italy, France, Belgium, Germany, Hungry, Austria, Poland, Czech Republic, Romania, Bosnia, Bulgaria, Croatia, Greece, Cyprus, Malta, Slovenia, Slovakia, Estonia, Latvia or Lithuania); (8) award of a certificate recognised, at minimum, at the national level. Training programmes aimed at other professional profiles such as psychologists, social workers, pharmacists, engineers or architects were excluded.

### 2.2. Search Strategy

Public and private universities were listed by country. The search engines on the university websites were then used to identify potential EPDMs by entering the following keywords: ‘disaster’, ‘catastrophes’ or ‘crisis intervention’. Additional records were identified through other sources (master’s degree advertising or by searching a specific search engine https://www.masterstudies.com/ (accessed on 18 November 2020).

### 2.3. Data Extraction

The search was conducted between June 2020 and April 2021. A PRISMA flow diagram was used to document the output of our search results (Figure 1) [24]. 

A total of 294 programmes were identified by two independent reviewers. The full information of 37 curricula were thoroughly screened and assessed for eligibility. Any disagreements between the researchers were discussed with a third reviewer. We finally included 34 programmes. The outcomes were: country, university and programme title; qualification type (master’s degree/postgraduate diploma or specialisation course); number of European Credit Transfer and Accumulation System (ECTS) credits awarded; programme duration; learning delivery modality (on-campus/online/blended); participation of teaching staff external to the university (yes/no) and their professional category; use of situational simulations (yes/no); use of information and communication technologies (ICTs; yes/no; which one); content included in the curricula; competencies acquired by the students.

### 2.4. Data Synthesis

Three of the researchers, who were all experienced female nurses with doctorate degrees working as assistant professors, conducted the data synthesis. They were guided by the principles of integrative analysis, which required them to order, encode, categorise and summarise the data. The overarching stages specified by Whittemore and Knafl [22], namely data reduction, data display and conclusion drawing, were followed. Qualitative analysis of the course curricula was conducted by content analysis to break down the data and group them into categories [25]. A descriptive analysis of the response frequencies was performed for each of the categorical items, and the mean and standard deviation were calculated (ẋ ± *SD*) for the continuous variables. SPSS^®^ statistical software (version 26.0) for Windows^®^ (IBM Corp., Armonk, NY, USA) was used to analyse the data.

## 3. Results

As summarised in Table 1, we identified 34 EPDMs from Belgium [26], Denmark [27,28], France [29,30,31,32,33], Germany [34,35,36], Greece [37], Italy [38,39], Portugal [40,41], Spain [42,43,44,45,46,47,48,49,50,51,52], Switzerland [53] and the United Kingdom [54,55,56,57,58,59]; 17 countries examined did not offer any relevant programmes. A total of 69.7% (*n* = 23) of these EPDMs were master’s degrees and 20.5% (*n* = 7) were specialisation courses. Some 89% of the programmes used the ECTS credit system, awarding a mean of 83.3 ECTS points (*SD* = 68.3; range 3–315 ECTS credits); however, no information on the academic credit system was available for 11% of the courses [31,32,35,56]. The programmes lasted from 2 weeks to 4 years, with those lasting one year being the most common (50%, *n* = 17). Only three EPDMs (8.8%) lasted less than 100 h [31,32,41].

The on-campus modality was used in 58.8% (*n* = 20) of cases; 17.6% (*n* = 6) were online and the remaining courses employed the blended modality. Virtual university classrooms were the most common ICT types used (87.9%, *n* = 29). Half of the programmes used situational simulations (*n* = 17), although no information on these was available for 27.3% of the programmes. Almost a third (*n* = 10) of the programmes used external multidisciplinary teachers such as health professionals, firefighters or civil protection members. The contribution of international organisations, such as the World Health Organization and non-governmental organisations including Doctors Without Borders and the Red Cross, in some programmes (the Piemonte Orientale programme) was of special note. Finally, despite the variability in the denominations of the different study programme subjects, there were ten main subcategories, each framed within the four phases described by Blanchard [60] for disaster management (Table 2). It should be noted that the information contained on the web pages of some EPDMs was scarce [30,32,33,37,50], while others were extensive and detailed [27,34].

Almost half of the programs addressed general concepts related to disasters: (i) risks and characteristics, (ii) types of disasters, (iii) disaster management and planning, (iv) professional intervention and (v) specific health training. Approximately, one-third of the programs included the ethical and legal framework, as well as psychological support [35,38,39,47,51,53] and international cooperation [28,34,38,42,43,45,47,51,52,58].

Few programs addressed specific content on the response to terrorist attacks [38], the management of refugees and camps in disasters [28] or pandemics [29,53]. Only two programs contemplated the development of intercultural cooperation [36,56], and none referred to the development of interprofessional competencies. We did not find any pattern to the contents of the programmes, according to the different countries.

## 4. Discussion

This study provides a general overview and an updated database of the European EPDM courses directed towards healthcare professionals that were available in the 2020/21 academic year. We aimed to identify whether universities offering EPDMs have improved the quality of their curricula in recent years [61] by incorporating the proposed recommendations that holistic and standardised programmes should have developed [15]. According to our results, there were 34 EPDMs based in Europe that were aimed at healthcare professionals. This figure is far from the 140 found by the DITAC group [15] because these researchers included courses aimed at any participant type. However, when the DITAC results were broken down, they had indeed included a similar number of programmes aimed at healthcare professionals.

Over the last 7 years, the geographical distribution of these courses remained stable, although the number of countries offering this type of programme had decreased. In line with international recommendations [7,62], most of the EPDMs were master’s degrees with an average duration of one year. Compared to the DITAC project, an increasing number of programmes used the ECTS credit system, although it is important to note that DITAC included professional courses undertaken in non-university settings (which did not use the ECTS system).

Although the most common learning modality was still in person, the COVID-19 pandemic had encouraged faculties to find new ways to improve student experiences and to ensure the safety of both students and staff. According to our results, the online modality increased from 11% to 18% in recent years. However, the blended modality was presented as the most useful and suitable teaching method for disaster management because it can combine both theory and practice as well as cooperative and individual work. Moreover, although still limited, thanks to the development of virtual teaching platforms, clinical simulations could be implemented without students having to attend every session in person [16,63]. These semi-presential courses allowed students to access course materials, online sessions and collaborative work rooms.

Computer-generated simulation (virtual reality or augmented reality) is a growing trend in the educational health context [64,65]. Virtual reality generates a three-dimensional image or environment that a person can interact with in a seemingly real or physical way using special electronic equipment, such as a helmet with a screen inside or gloves fitted with sensors. Augmented reality is an enhanced version of the real physical world that is achieved through digital visual elements, sound or other sensory stimuli delivered via technology. Moreover, the augmented reality technology superimposes a computer-generated image on a user’s view of the real world, thus providing a composite view.

The computer-generated simulation could be incorporated into EPDMs to allow students to train for different scenarios by recreating situations and psychological states as if they were real [64]. Students could interact with different in-disaster or post-disaster environments without putting themselves or third parties in danger. For example, they could train their risk management of geological processes such as earthquakes, tsunamis or floods [65].

In line with expert recommendations, and following the aforementioned teaching methodologies, half of the identified programmes used clinical simulations [8,11]. Simulation is considered the most effective teaching–learning methodology currently available [63,66] for training complex scenarios (for example, triaging multiple victims or hospital situations in the field). This self-directed learning combines theory and practice, engages students and increases their levels of satisfaction and self-confidence [67,68,69]. However, as with all virtual reality systems, high-fidelity simulation has a huge economic cost because it requires the use of standardised mannequins or patients, software packages and specific training for instructors [64,69].

In another vein, we noted that the main subjects addressed in the EPDMs reviewed went beyond those defined by DITAC [15] (management, vulnerability analysis, logistics and transport, law and ethics, and protection and security). In addition to these topics, the EPDMs considered the management of communications, interventions and evaluations, epidemiology and biostatistics, and post-disaster responses, and, in the case of master’s degrees, also incorporated a dissertation. The end-of-course dissertation is a relevant way to develop future researchers’ skills to solve clinical practice problems.

Although the ‘response phase’ [60] has been highlighted as a desirable part of the course content [10,70,71], the university websites reviewed contained insufficient data. For example, we extracted information from just one EPDM about how to address terrorist events, even though the academic literature suggests inadequate levels of preparedness among healthcare staff in this regard [8]. Recent terrorism events in Europe have shown the importance of being prepared, establishing community coalitions in advance to promote efficient and effective mobilization, and responding successfully to the mental and physical health needs of individuals affected by such disasters [72].

Similarly, the management of pandemics was named by just two EPDM. SARS-CoV-2 has been considered a pandemic since 11 March 2020. With almost 200 million confirmed global cases of COVID-19, more than 4 million resulting deaths and in excess of 3500 million vaccine doses administered, the pandemic still continues to spread [73]. Thus, in our opinion, this highlights the fact that specific subject areas related to significant emergencies such as terrorism or the management of pandemics must be included in EPDMs to train professionals to deal with the severe psychological and physical consequences of such acute emergency events. We suggest that EPDMs should be regularly updated to include diverse types of disasters and that these education programmes must be made consistent with the national plans and healthcare systems of different geographical regions [8].

Finally, it is important to mention the general structure of the revised training programmes. DITAC proposed a comprehensive programme split into two cycles: the first in an undergraduate setting, and the second in the context of a master’s degree, each worth 30 ECTS credits. However, their intended purpose was not to train course members in their profession or area of specialisation (that is, instructing nurses in nursing or logisticians in logistics) but, rather, education in the specific elements of disaster management [10]. However, this proposal contradicts the general perception of doctors and nurses about their deficit in technical competence in the context of disaster management [16,17,18,19,20], a perception that arises from the lack of undergraduate degree training in subjects related to emergencies at most European universities.

We cannot aspire to create a revolution in the undergraduate teaching plans of potential disaster responders in the short to medium term. Thus, we think a more attainable goal would be the implementation of a first cycle worth 30 postgraduate ECTS for the acquisition of knowledge in order to train doctors and nurses.

This would then be complemented by a second cycle, also worth 30 ECTS, to improve their performance in terms of intercultural cooperation (how to work and behave in an international team), intercultural awareness (understanding professional skills in different intercultural contexts) and interprofessional skills (practical training with an interinstitutional approach, especially in communication skills and flexibility) [10].

Although our results indicate that the current EPDM course coverage of the topics from the first cycle is adequate, only two EPDMs seem to have, as of yet, incorporated the skills suggested for the second cycle. This deficit was also noted when considering that only one-third of the programmes incorporated non-university-based multidisciplinary teaching teams to facilitate closer descriptions of the realities of disasters [10,62,74], an aspect that should be generally improved by EPDM managers. As a summary, to provide guidance for updates or new potential programmes aimed at healthcare professionals, we have provided evidence-based suggestions in Figure 2.

### Limitations

The first limitation of this current work was its inclusion criteria. The EPDM search was limited to university contexts only, even though this type of accreditation is obtained by other means in some regions of the world. However, the 2015–30 Sendai Framework placed human health at the centre of disaster management and considers the development of healthcare personnel in this area a vital issue [6]. This paper wanted to provide updated data on what EPDMs look like, and how they can be better reorganized to improve healthcare personnel competencies. Moreover, all of the authors are professors at a European university that teaches an EPDM course directed towards doctors and nurses.

Another possible limitation was the methodology we used to search for training programmes. Although we performed an in-depth search, some relevant training courses may have been missed. As Ingrassia et al. [63] identified, this is probably because there is no comprehensive database for training and educational initiatives for disaster management, either at the European or national level.

It is necessary to provide complete and detailed information on EPDMs on the web pages and use a standardized vocabulary based on the literature. It is also necessary that the extensive offer of EPDMs is compiled in a single institutional or collegial platform. Different European associations, organizations or institutions interested in disaster risk reduction or management could favour this compilation of training offers, such as the Emergency Response Coordination Centre (ERCC) for European Civil Protection and Humanitarian Aid Operations [75]. The Centre ensures the cooperation and coherence of European Union action at the interinstitutional level, focusing on coordination mechanisms with the European External Action Service, the Council and European Union Member States. It also acts as a permanently available contact point when the solidarity clause is invoked and provides emergency communications and monitoring tools through the Common Emergency Communication and Information System (CECIS), an alert and notification application to exchange information in real time. There are other non-European organizations with an interest in disaster risk reduction or management. For example, the Alliance of International Science Organizations on Disaster Risk Reduction (ANSO-DRR) is an international, non-profit and non-governmental scientific alliance bringing together academies of science, research organizations and universities that share a strong interest in disaster risk reduction in the regions along the land-based and maritime routes of the Belt and Road Initiative [7].

Nonetheless, this exploratory work did provide useful information that could be further expanded by consulting critical informants to obtain information that may not be otherwise readily obtainable and to identify new programme topics. Finally, the lack of similar studies in other regions meant that there were no benchmarks with which to compare our results or to monitor the evolution of this type of education globally over time. For future recommendations, similar studies should be conducted to enable the benchmarking of programmes.

## 5. Conclusions

This study presents an updated database of the EPDMs available at European universities in the 2020/21 academic year and provides educators and researchers in healthcare with an understanding of the current state of the training in health emergency and disaster management. EPDMs have improved the quality of their teaching methodologies by making them more practical and interactive, introducing relevant topics such as communication and the evaluation of interventions, and developing research skills through dissertations. However, the EPDMs have not yet incorporated the recommendation about intercultural and interprofessional awareness skills.

## Figures and Tables

**Figure 1 ijerph-18-11455-f001:**
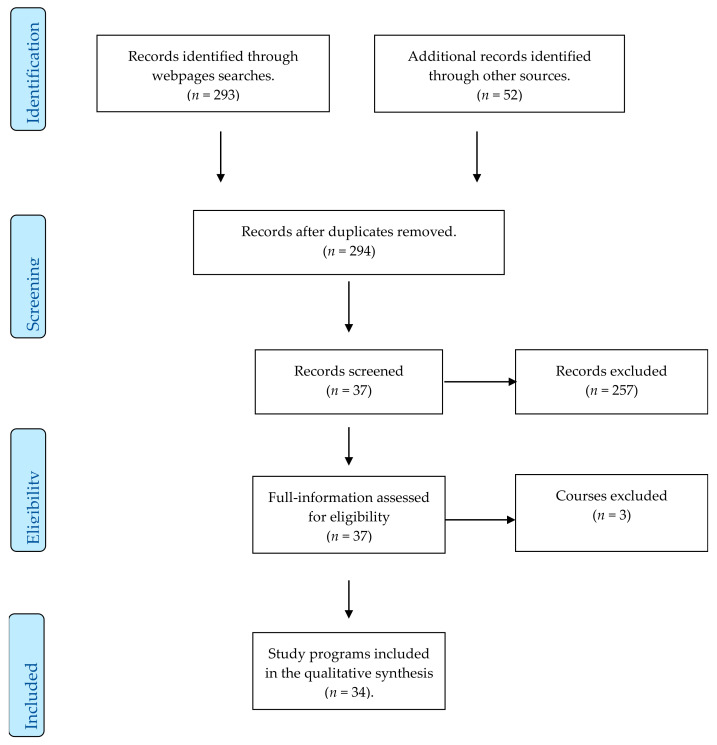
PRISMA flow diagram of the search process.

**Figure 2 ijerph-18-11455-f002:**
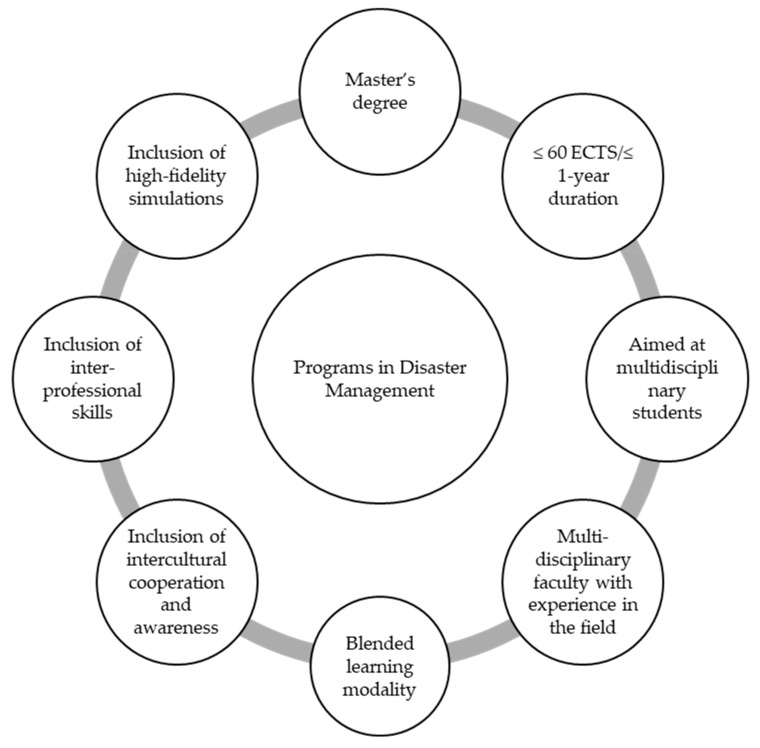
Suggestions to establish and improve the curriculums used in disaster management training.

**Table 1 ijerph-18-11455-t001:** Characteristics of the educational programmes in disaster management assessed in this study.

Title	Country and University	Qualification Awarded, ECTS/Duration	Delivery Modality	Simulation Training Method	External Teaching Staff	ICTs
Risk and Disaster Management [26]	Belgium, University Sciences of Liège	Master’s degree, 60 ECTS/1 year	On-campus	Unknown	Yes	Virtual classroom
Disaster Management [27]	Denmark, University of Borne	Master’s degree, 60 ECTS/1 year	On-campus	Yes	No	Virtual classroom
Disaster Management [28]	Denmark, University of Copenhagen	Master’s degree, 60 ECTS/1 year	On-campus	Yes	Unknown	No
Environment, Health, and Disaster Management [29]	France, University of Montreal	Postgraduate diploma, 30 ECTS/1 year	On-campus	No	No	Virtual classroom
Nursing in Disaster Situations [30]	France, University of Neith	Specialisation course, 20 ECTS/1 year	On-campus	Yes	No	No
Disaster Medicine and Exceptional Health Situations [31]	France, University of Rennes	Postgraduate diploma, ECTS unknown/96 h	On-campus	Unknown	No	Virtual classroom
Disaster Medicine [32]	France, University of Lorraine	Postgraduate diploma, ECTS unknown/96 h	On-campus	Unknown	Unknown	Virtual classroom
Disaster and Natural Risk Management [33]	France, University of Paul Valéry	Master’s degree, 60 ECTS/1 year	On-campus	No	No	Virtual classroom
Disaster Management and Risk Governance [34]	Germany, University of Bonn	Master’s degree, 120 ECTS/3 years	On-campus	Yes	Civil Protection	Virtual classroom
Emergency Management [35]	Germany, University of Hochschule	Master’s degree, ECTS unknown/4 years	Blended	Yes	Unknown	Virtual classroom
Crisis and Emergency Management [36]	Germany, Carl Remigius Medical School	Master’s degree, 90 ECTS/2 years	Blended	Yes	Unknown	Virtual classroom
World Health—Disaster Medicine [37]	Greece, University of Athens	Master’s degree, 120 ECTS/2 years	On-campus	Unknown	EMs, academics, military, civil protection unit and healthcare organisations	Unknown
Crisis, Emergency, and Disaster Management [38]	Italy, University of Ciels	Master’s degree, 120 ECTS/2 years	On-campus	Yes	No	Virtual classroom
European Disaster Medical Sciences [39]	Italy, University of Piemonte Orientale	Master’s degree, 60 ECTS/1 year	On-campus/Online	No	WHO members, Doctors Without Borders NGO, European Society of EMs.	Virtual classroom
Emergencies, Traumas, and Disasters [40]	Portugal, Santa Maria School of Health	Master’s degree, 30 ECTS/1 year	Unknown	Unknown	Unknown	Virtual classroom
Emergency and Disaster Medicine [41]	Portugal, University of Nova	Specialisation course, 3 ECTS/2 weeks	On-campus	Unknown	Unknown	Unknown
Emergencies and Catastrophes [42]	Spain, University of Alicante	Master’s degree, 60 ECTS/1 year	Blended	Yes	Yes	Virtual classroom
Study of Interventions in Emergencies, Disasters, and International Cooperation [43]	Spain, University of Camilo José Cela	Master’s degree, 60 ECTS/1 year	Online	Yes	No	Virtual classroom
Disaster Management [44]	Spain, University CEU-Cardenal Herrera	Specialisation course, 16 ECTS/400 h	Online	Yes	No	Virtual classroom
Disaster Management [45]	Spain, Complutense University of Madrid	Master’s degree, 90 ECTS/2 years	On-campus	Yes	Unknown	Virtual classroom
Security, Crisis and Emergency Management [46]	Spain, University of Rey Juan Carlos	Master’s degree, 60 ECTS/1 year	On-campus	Yes	Unknown	Virtual classroom
Integrated Disaster Risk Management [47]	Spain, University of Rey Juan Carlos	Specialisation course, 30 ECTS/300 h	On-campus	Yes	Yes	Virtual classroom
Emergencies and Disasters [48]	Spain, University San Pablo CEU	Master’s degree, 60 ECTS/1 year	On-campus	Yes	Yes	Virtual classroom
Emergencies and Disaster Analysis and Management [49]	Spain, University of Oviedo	Master’s degree, 60 ECTS/1 year	On-campus	Unknown	Unknown	Virtual classroom
Emergency Health Care in Extreme Situations and Disasters [50]	Spain, Alcala Formation	Specialisation course, 20 ECTS/500 h	Online	No	No	Virtual classroom
Integrated Care in Health Catastrophes [51]	Spain, Alcala Formation	Specialisation course, 8 ECTS/200 h	Online	No	No	Virtual classroom
Civil Strife, Disasters, and Catastrophes [52]	Spain, Escuela SAMU	Specialisation course, 29 ECTS/725 h	Blended	Unknown	Unknown	Unknown
Public Health Disaster Management [53]	Switzerland, James Lind Institute	Master’s degree, 138 ECTS/2 years	Online	Unknown	Unknown	Virtual classroom
Disaster Management [54]	United Kingdom, University of Bournemouth	Master’s degree, 180 ECTS/1–2 years	On-campus/Online	Unknown	Unknown	Spreadsheets and word processing
Disaster Management and Resilience [55]	United Kingdom, University of Coventry	Master’s degree, 180 ECTS/1–3 years	On-campus/Online	Yes	Unknown	Virtual classroom
Risk, Crisis and Disaster Management [56]	United Kingdom, University of Leicester	Master’s degree ECTS unknown/2 years	Online	No	Safety and Civil Protection Unit	Virtual classroom and Digital Library
Crisis and Disaster Management [57]	United Kingdom, University of Lincoln	Master’s degree, 315 ECTS/1–2 years	On-campus	Yes	Civil Protection Unit	Virtual classroom
International Disaster Management [58]	United Kingdom, University of Manchester	Master’s degree, 180 ECTS/1–2 years	On-campus	No	NGOs and civil organisations	Virtual classroom
Risk, Crisis and Resilience Management [59]	United Kingdom, University of Portsmouth	Master’s degree, 180 ECTS/1–2 years	On-campus	Yes	Fire and Rescue Service and City Council	Spreadsheets and word processing

ICTs, information and communication technologies; ECTS, European Credit Transfer and Accumulation System; NGOs, non-governmental organisations; EMs, emergency medical staff.

**Table 2 ijerph-18-11455-t002:** Categories and Subcategories of Disaster Management, framed within Blanchard’s four phases of disaster management [60].

Categories	Subcategories
**Phase 1. Preparedness**	**Introduction to risks and disasters** [26,27,28,29,30,31,33,34,35,39,44,47,51,53,59]. These standard curricula subjects provided knowledge about disasters and their risks. They included the general concepts, characteristics of risks and population vulnerability analyses.
	**Typology of risks and disasters** [27,28,29,31,33,34,37,38,40,42,43,45,47,49,52,53,54,56]. The relevance varied among the programmes from ‘curriculum subject’ to ‘topic’. These included classification of risk and disasters, risk estimation and the hazard–threat binomial.
	**Legal framework** [27,28,29,31,34,36,39,42,43,46,49] This included legal topics related to disaster and emergency management, the functions of governments and general and specific national and international regulations.
**Phase 2. Response**	**Disaster management and planning** [26,27,28,29,30,31,32,33,34,35,36,38,39,40,41,42,43,45,46,47,49,51,53,54,55,56,57,58,59] This included management stages, management strategies and techniques, leadership, management to reduce risk (security), logistics and medical management.
	**Communication management** [27,29,31,34,35,40,46,47,51,53,54] These curricula subjects explained how organisations, countries and people involved in a disaster communicate and stay updated at all times.
	**Intervention and evaluation** [27,30,34,38,40,41,42,43,44,45,46,47,49,54,55,56,57,58] This included specific knowledge about the professional response, its analysis and evaluation using different tools and feedback.
	**Epidemiology and biostatistics** [27,29,34,52,53,58] This standard curriculum subject provided knowledge for the collection, analysis and understanding of data.
	**Specific health training in disasters** [27,30,31,32,34,36,37,39,40,41,42,43,44,45,48,50,51,52] A standard curricula subject that provided general knowledge about prevalent pathologies according to the disaster type and that trained specific healthcare team skills.
**Phase 3–4. Recovery and Mitigation**	**Post-disaster response** [26,28,35,38,39,45,47,51,53,54,58,59] These standard curricula subjects included the impact of disasters on populations and communities, crisis intervention, emotional recovery and post-traumatic stress disorder.
**Dissertation**	Master’s programmes included a final assignment that allowed the students to demonstrate the integration of the training contents and their skills/capabilities [26,27,33,34,42,43,45,46,47,48,49,50,53,56,58,59].

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
