# Peer review of "European Educational Programmes in Health Emergency and Disaster Management: An Integrative Review"

_ijerph, 2021, doi:10.3390/ijerph182111455_

Round 1

Reviewer 1 Report

I was invited to revise the paper entitled "European educational programmes in health emergency and disaster management: an integrative ". It aimed to review characteristics educational programmes in disaster management in the european region. The topic is interesting and this review can improve the knowledge in this field.

Major observation

In my opinion Authors should discuss differences in topic and subjects proposed by different courses. In addition, should be described differences across countries.

Minor observation

The title should be rephrased.

Author Response

We send our response to the comments and resubmit our manuscript ID IJERPH-1409779 entitled " European educational programmes in health emergency and disaster management: an integrative review."

The researchers are grateful for the issues raised and would like to ensure that the changes made to the manuscript meet the journal's criteria. We have introduced or corrected all points. Please find below a table summarising the work done

Reviewer 2 Report

Thank you for the opportunity to review this manuscript entitled, "European educational programmes in health emergency and disaster management: an integrative review". This document is well-written and easy to follow.

The authors did an excellent job and with minor changes this manuscript can be accepted for publication.

Add some suggestions for future research.

Please refer to the attached document with the relevant in-text comments. 

Author Response

(The authors gave the same response as above.)

Reviewer 3 Report

This paper analyzes the content characteristics and teaching methods of health emergency and disaster management education programs in 34 universities in Europe. Although the academic nature of the article is not strong enough and the content of the analysis is not comprehensive and in-depth, the author has done a lot of analysis, and the article is of great significance for the cultivation of emergency management personnel's literacy. Therefore, in my opinion, this manuscript can be accepted, if the author can revise the following suggestions:

  1. Figure 1 is not clear, and Figure 2 is missing the title, please modify it.
  2. Lines 165 and 166 is the same.

Author Response

We send our response to the comments and resubmit our manuscript ID IJERPH-1409779 entitled " European educational programmes in health emergency and disaster management: an integrative review."

The researchers are grateful for the issues raised and would like to ensure that the changes made to the manuscript meet the journal's criteria. We have introduced or corrected all points. Please find below a table summarising the work done.

Round 2

Reviewer 1 Report

Authors addressed all points. Paper is now acceptable for publication.

Author Response

Thank you for your comments.

Reviewer 2 Report

Thank you for the opportunity to comment on the revised manuscript. In general, it reads much better now. The revision has been done sufficiently and the manuscript can be accepted with some minor revisions:

  1. Some minor typos were detected: p5of19 lines 294-297: Please relook at the sentence stated below as (i) it is difficult to follow. Consider splitting it to two sentences. (ii)  is the 6 at the end of the sentence a reference or not? Please change accordingly.

    "On the other hand, our main motivation was to provide data about how EPDMs can be best updated in these health professionals,  in line with the 2015–30 Sendai Framework placed human health at the centre of disaster management and considers the development of healthcare personnel in this area a vital issue6."

    - it is difficult to follow. Consider splitting it to two sentences. - is the 6 at the end of the sentence a reference or not?

Author Response

(i) we have split it into two sentences. "However, the 2015–30 Sendai Framework placed human health at the centre of disaster management and considers the development of healthcare personnel in this area a vital issue6. This paper wanted to provide updated data on what EPDMs look like and how they can be better reorganized to improve healthcare personnel competencies."

(ii) Typo error amended. Yes, at the end of the sentence the 6 is a reference. We have changed the format to superindex, according to the rest of the references in the text.